# D,L-Lysine-Acetylsalicylate + Glycine (LASAG) Reduces SARS-CoV-2 Replication and Shows an Additive Effect with Remdesivir

**DOI:** 10.3390/ijms23136880

**Published:** 2022-06-21

**Authors:** Johannes Jungwirth, Clio Häring, Sarah König, Liane Giebeler, Heena Doshi, Christian Brandt, Stefanie Deinhardt-Emmer, Bettina Löffler, Christina Ehrhardt

**Affiliations:** 1Section of Experimental Virology, Institute of Medical Microbiology, Center for Molecular Biomedicine (CMB), Jena University Hospital, Hans-Knoell-Str. 2, D-07745 Jena, Germany; johannes.jungwirth@med.uni-jena.de (J.J.); clio.haering@med.uni-jena.de (C.H.); sarah.koenig2@med.uni-jena.de (S.K.); liane.giebeler@med.uni-jena.de (L.G.); 2Institute of Molecular Cell Biology, Center for Molecular Biomedicine (CMB), Jena University Hospital, Hans-Knoell-Str. 2, D-07745 Jena, Germany; heena.doshi@med.uni-jena.de; 3Institute for Infectious Diseases and Infection Control, Jena University Hospital, Am Klinikum 1, D-07747 Jena, Germany; christian.brandt@med.uni-jena.de; 4Institute of Medical Microbiology, Jena University Hospital, Am Klinikum 1, D-07747 Jena, Germany; stefanie.deinhardt-emmer@med.uni-jena.de (S.D.-E.); bettina.loeffler@med.uni-jena.de (B.L.)

**Keywords:** SARS-CoV-2, COVID-19, LASAG, pro-inflammation, remdesivir, chip model

## Abstract

The severe acute respiratory syndrome coronavirus 2 (SARS-CoV-2) causing the coronavirus disease-19 (COVID-19) is still challenging healthcare systems and societies worldwide. While vaccines are available, therapeutic strategies are developing and need to be adapted to each patient. Many clinical approaches focus on the repurposing of approved therapeutics against other diseases. However, the efficacy of these compounds on viral infection or even harmful secondary effects in the context of SARS-CoV-2 infection are sparsely investigated. Similarly, adverse effects of commonly used therapeutics against lifestyle diseases have not been studied in detail. Using mono cell culture systems and a more complex chip model, we investigated the effects of the acetylsalicylic acid (ASA) salt D,L-lysine-acetylsalicylate + glycine (LASAG) on SARS-CoV-2 infection in vitro. ASA is commonly known as Aspirin^®^ and is one of the most frequently used medications worldwide. Our data indicate an inhibitory effect of LASAG on SARS-CoV-2 replication and SARS-CoV-2-induced expression of pro-inflammatory cytokines and coagulation factors. Remarkably, our data point to an additive effect of the combination of LASAG and the antiviral acting drug remdesivir on SARS-CoV-2 replication in vitro.

## 1. Introduction

The severe acute respiratory syndrome coronavirus 2 (SARS-CoV-2), that emerged in late 2019, is the pathogen causing COVID-19 [1]. The virus has rapidly spread around the world, leading to a pandemic with more than 500 million cases worldwide and more than 6 million deaths (updated information from the World Health Organization, as of 9 June 2022). In the meantime, frequent mutations have resulted in multiple variants of SARS-CoV-2, whose specific sequences cause changes in the viral properties [2].

Although infection is mild or asymptomatic in many cases, SARS-CoV-2 can cause severe courses including life-threatening pneumonias with a high mortality rate [1]. In particular, older patients with comorbidities, such as lung pathologies, cardiovascular diseases, diabetes predisposition and obesity often develop severe illnesses and complications [3,4,5]. The early stages of SARS-CoV-2 infection are characterized by a high virus load and milder symptoms—most commonly fever and dry cough [6]. With the progression of the disease, the lungs become increasingly affected. As the viral load in the lungs decreases, hyperactivation of the host immune response can result in overwhelming problems. This can ultimately lead to a cytokine storm or sepsis and death [7,8]. Additionally, COVID-19 patients are reported to have an increased risk of venous thromboembolism, arterial thrombosis and bleeding complications [9,10,11] caused by hemostasis disorders and abnormal blood coagulation parameters [12,13].

Although vaccination campaigns are quickly progressing in many countries, there is still an urgent need for effective antiviral therapies because of low coverage and less protection against new emerging virus variants. For this reason, the repurposing of existing drugs that have already proven to be safe in the treatment of other diseases represents a promising alternative [14,15]. Among those drugs are antivirals targeting the virus itself, medications that interfere with immune responses (either by inhibiting overwhelming cytokine production or by supporting cellular antiviral defense mechanisms) as well as substances that regulate virus-supportive cellular functions [16]. There are now numerous therapy recommendations that initially pursue an antiviral strategy and, when viral load has subsided, an anti-inflammatory strategy [17]. Among the antiviral agents, Veklury^®^ (remdesivir, Gilead Sciences), that was originally developed for the treatment of Ebola and was quickly repurposed for the treatment of COVID-19 patients, is important [18,19]. Furthermore, the antiviral Lagevrio^®^ (molnupiravir, Merck Sharp & Dohme) [20] as well as the RNA-polymerase inhibitor Avigan^®^ (favipiravir, Fujifilm) [21,22] are in use. Recently the protease inhibitor Paxlovid^®^ (nirmatrelvir/ritonavir, Pfizer) was approved [23]. Frequently used anti-inflammatory substances are glucocorticoids [24] and the interleukin-6 antagonist tocilizumab [25,26], even if newer studies show almost no positive outcome [16]. Among the monoclonal antibodies that target the virus, Ronapreve^®^ (casirivimab/imdevimab, Roche) [27] and Regkirona^®^ (regdanvimab, Celltrion) [28] have to be mentioned.

The underlying molecular mechanisms regulating pathophysiology during SARS-CoV-2 infections in the context of therapeutic treatment are not clear in detail. Likewise, the side effects of commonly used pharmaceutical substances to combat lifestyle diseases, such as high blood pressure, arterial calcification, diabetes or to treat non-specific pain and inflammation are rarely investigated in the context of infection. Among these substances, acetylsalicylic acid (ASA), commercially available as Aspirin^®^ (Bayer), is the most commonly used drug for the treatment of pain and fever due to various causes [29]. ASA is widely used for its analgesic, anti-pyretic and anti-inflammatory effects in patients suffering from different diseases [30]. The anti-inflammatory and analgesic effects of ASA, similar to those of other salicylates, are related to its ability to inhibit nuclear factor kappa B (NFκB)-mediated signaling and cyclooxygenases (COX) [31,32]. 

A newer formulation of ASA, D,L-lysine-acetylsalicylate + glycine (LASAG), is a water-soluble salt of ASA. The two amino acids lysine and glycine improve the stability and tolerability of the active compound ASA (Appendix A) [31,33,34,35]. LASAG is approved as Aspirin^®^ i.v. 500 mg and can be applied orally, intravenously or via inhalation [31]. Remarkably, it has been shown to have antiviral effects against influenza A virus (IAV) infection in vitro, in a mouse model and in clinical studies in patients with severe influenza [31,36,37]. It has also been shown to impair the propagation of different coronaviruses in vitro including the highly pathogenic Middle East respiratory syndrome coronavirus (MERS-CoV) [34]. Patients who were already on low dose ASA due to cardiovascular diseases before the SARS-CoV-2 infection suffer from less severe COVID-19 and the use of ASA in hospitalized patients is associated with a lower risk of mechanical ventilation, intensive care unit (ICU) admission and in-hospital mortality [38]. ASA is now part of many treatment regimes of COVID-19 [39].

In the present work, we investigated beneficial and adverse effects of LASAG on SARS-CoV-2 replication and SARS-CoV-2-induced host cell response in simple and complex cell culture systems in vitro. Our data demonstrate an inhibitory effect of LASAG on SARS-CoV-2 infection and SARS-CoV-2-induced expression of pro-inflammatory cytokines as well as thrombotic factors. When LASAG was used in combination with remdesivir, an additive effect of the antiviral activity of both substances could be demonstrated in vitro.

## 2. Results

### 2.1. LASAG Inhibits SARS-CoV-2 Replication at Non-Toxic Concentrations In Vitro

The antiviral potential of LASAG was already demonstrated on human coronaviruses and MERS coronavirus in vitro [34]. 

Here, the effective concentration of 50% (EC_50_) of LASAG was determined by standard plaque assay 24 h post infection (p.i.) with a SARS-CoV-2 Alpha variant (isolate 5159) and a Delta variant (isolate 4749). LASAG treatment resulted in an efficient inhibition of the virus replication at millimolar concentrations (Alpha variant: EC_50_ of 6.7 mM; Delta variant: EC_50_ of 5.4 mM) (Figure 1a,b). To exclude the notion that the described effects of LASAG on the viral replication are due to impaired cell viability, cell propagation was analyzed. For that, Calu-3 and Vero-76 cells were counted 24 h after treatment with LASAG (Figure 1c,d). The results clearly show that the used concentrations of LASAG do not affect cellular proliferation and viability at all in Calu-3 cells and only slightly in Vero-76 cells.

### 2.2. In the Presence of LASAG, the Replication of Different SARS-CoV-2 Variants Is Reduced In Vitro

To further elucidate the effect of LASAG on the replication of three different SARS-CoV-2 Alpha variants and one Delta variant, we used the human lung carcinoma cell line Calu-3 and primate kidney Vero-76 cells. Here, progeny infectious virus particles as well as viral protein expression were examined at 24 h p.i. The data indicate a significant reduction in SARS-CoV-2 titers in the samples treated with 10 mM LASAG (Figure 2a–e left panels). Consistently, SARS-CoV-2 spike protein expression of both units of the spike 2 subunit was reduced in the presence of LASAG as visible in Western blot (Figure 2a–e middle panels) and densitometric analyses (Figure 2a–e right panels). Furthermore, LASAG-mediated effects on SARS-CoV-2 protein expression were verified by immunofluorescence (IF) studies (Figure 2f).

Thus, the treatment of SARS-CoV-2-infected epithelial cells with LASAG at non-toxic concentrations efficiently reduced viral replication and propagation.

### 2.3. LASAG Reduces SARS-CoV-2-Induced Cytokine and Chemokine Production In Vitro

The cytokine storm that is induced in severe cases of COVID-19 is known to be a key cause of morbidity in SARS-CoV-2 infection [40,41]. One major goal of this work was to investigate adverse side effects of LASAG during SARS-CoV-2 infection, especially since LASAG is commonly used to combat inflammation [42]. Initially, the effect of LASAG on viral mRNA expression was determined by qRT-PCR analyses 24 h p.i. (Figure 3a). Consistent with the data on viral titers and viral protein expression (Figure 2a–d), a significantly reduced mRNA synthesis of SARS-CoV-2 (N1) was observed in Calu-3 cells treated with 10 mM LASAG. Concomitantly, the SARS-CoV-2-mediated interferon-gamma-induced protein 10 kDa (IP-10), the interferons (IFNs) IFNβ, IFNλ1 and IFNλ2/3, the tumor necrosis factor-related apoptosis-inducing ligand (TRAIL) and the 2′-5’-oligoadenylate synthetase 1 (OAS1) mRNA synthesis were significantly reduced upon LASAG treatment (Figure 3a) probably due to reduced viral infection.

Furthermore, the reduction in SARS-CoV-2-induced cytokine expression was confirmed by flow cytometry (LEGENDplex^TM^) analyses. Here, supernatants of uninfected or SARS-CoV-2-infected, untreated or LASAG-treated Calu-3 cells were examined. At 24 h p.i. with SARS-CoV-2, a significant induction of interleukin-6 (IL-6), IP-10, IFNλ1, IFNλ2/3, IFNβ, IFNγ, IL-10 and tumor necrosis factor α (TNFα) protein secretion was observed (Figure 3b). LASAG treatment resulted in reduced levels of the SARS-CoV-2-induced inflammatory factors except IL-6. Another hallmark of severe COVID-19 is the dysregulation of hemostasis. Even though endothelial tissue plays a pivotal role in hemostasis, the lung epithelial cells were also shown to regulate coagulopathies [43]. The analysis of hemostatic factors showed a LASAG-dependent significant reduction in plasminogen activator inhibitor-1 (PAI-1) but almost no changes in tissue-type plasminogen activator (tPA) 24 h p.i. of Calu-3 cells (Figure 3c). Viral protein expression was already observed at 8 h p.i. in Western blot analysis and immunofluorescence studies and was reduced in the presence of LASAG (Appendix A), accompanied by an inhibition of SARS-CoV-2-induced STAT1 phosphorylation (Appendix A). Furthermore, cytokine expression was analyzed by flow cytometry. However, the results show no SARS-CoV-2-mediated induction of cytokines at this time point of infection (Appendix A).

### 2.4. LASAG Reduces Cytokine and Chemokine Production of SARS-CoV-2-Infected Epithelial Cells within the Chip Model

To investigate the effects of LASAG on SARS-CoV-2 infection in a more complex cell culture model, the previously described chip model [44] containing epithelial Calu-3 cells, endothelial human umbilical vein endothelial cells (HUVEC) and integrated peripheral blood mononuclear cells (PBMC) was used. Analogue to the experiments described above, both cellular layers were pre-treated with LASAG for 1 h. Furthermore, LASAG was added during and after the infection. As expected from previous experiments, SARS-CoV-2 only infects the epithelial layer of the chip model (Figure 4a), but not the endothelial layer (Appendix A), which we confirmed by the visualization of SARS-CoV-2 spike protein expression 28 h p.i. in immunofluorescence microscopy studies. The SARS-CoV-2 infection of the epithelial layer was reduced in the presence of LASAG (Figure 4a).

The infection resulted in a slight induction of IL-6, IP-10, IFNλ1, IFNλ2/3 and IFNβ protein secretion (Figure 4b) measured by flow cytometry of the supernatants of the epithelial chamber. Analogue to the in vitro results of mono cultured Calu-3 cells, the IP-10 and IFNλ1 levels were significantly reduced in the presence of LASAG in the chip model. The other SARS-CoV-2-induced cellular factors were only slightly reduced. The levels of IFNγ, IL-10 and TNFα were not affected at all (Figure 4b).

Since the endothelial tissue plays a pivotal role in hemostasis, we focused on the effect of LASAG on coagulation factors. Although the endothelial layer was not infected in the chip model, a slight activation of IL-6, IL-8 and tissue factor (TF) (Figure 4c) was demonstrated, probably due to virus-induced factors of the epithelial layer [44]. Of note, TF and tPA were significantly reduced in infected, LASAG-treated endothelial cells (Figure 4c). Other cytokines and thrombotic factors (IL-6, IL-8 and PAI-1) were slightly but not significantly downregulated in the presence of LASAG. All these factors are known to be aberrantly changed during SARS-CoV-2 infection.

It has to be mentioned that LASAG treatment of the endothelial layer in the chip model showed slight disturbance of the membrane function independent of SARS-CoV-2 infection (Appendix A), which was not observed in endothelial cells cultured in cell culture plates (Appendix A).

### 2.5. SARS-CoV-2 Replication Is Strongly Inhibited upon Combinatory Treatment with LASAG and Remdesivir, Which Shows an Additive Inhibitory Effect In Vitro

Since antiviral substances such as remdesivir are often used in the case of a SARS-CoV-2 infection, the combinatorial treatment of LASAG and remdesivir during SARS-CoV-2 infection of Calu-3 cells was investigated. Thus, cells were treated with LASAG–remdesivir combinations in different concentrations before, during and after infection. For all combinations, a potent reduction in virus titers (Figure 5a) and increasing inhibition of SARS-CoV-2 replication (Figure 5b) was detected, but no increase in toxicity (Figure 5c, Appendix A). To examine the pharmacological interactions of the drug pair, we used the three commonly used reference synergy models: Bliss independence, highest single agent (HSA) and zero interaction potency (ZIP) [45]. The results, shown in Appendix A, prove that combinations of LASAG and remdesivir were additive against SARS-CoV-2 infection in vitro. The lowest concentrations that show an additive inhibitory effect on SARS-CoV-2 replication were 2000 µM LASAG in combination with 0.1 µM remdesivir (Appendix A).

## 3. Discussion

ASA is one of the most commonly used drugs worldwide. It is used for many indications, mainly due to its impact against inflammation and pain of a general nature. In addition to these analgesic and anti-inflammatory properties, ASA has anti-platelet, anti-thrombotic and neuroprotective effects [46]. However, adverse effects of ASA (e.g., bleeding, Reye’s Syndrome in childcare) have to be taken into account, especially when high doses are administered [47,48].

In particular, several studies on the administration of ASA in COVID-19 patients show a variable picture on the outcome [38,39,48,49,50,51,52,53,54]. On the one hand, an improvement in mechanical ventilation and in-hospital deaths [38], and on the other hand, no effect of ASA on mortality [39,48,50,51,53,54] were reported. While ASA administration was associated with a decreased risk of thrombosis, an increased risk of bleeding occurred [48]. Although some patients who took ASA prior to hospital admission had less need for upgrading ventilatory support [39], other patients had an increased risk of death using ASA before COVID-19 [52].

Based on the fact that severe courses of COVID-19 are often associated with complications caused by inflammatory processes and blood clotting and that LASAG is used for many indications, the present study was intended to investigate beneficial and adverse effects of LASAG during SARS-CoV-2 infections in vitro.

Here, we were able to show that the replication of SARS-CoV-2 in vitro is reduced in the presence of 10 mM LASAG. Similar observations have been made previously for the coronaviruses, including MERS coronavirus but also IAV in vitro [34,36]. These studies indicated that the reduction in viral replication relied on the inhibition of virus-supportive functions of NFκB-mediated signaling during infection. Especially in the case of IAV infection, it was demonstrated that IAV-induced detrimental hyperinflammation was reduced upon NFκB inhibition [55]. Likewise, NFκB is implicated in the development of acute respiratory distress syndrome in COVID-19 patients due to the hyper-activation of the immune system [56,57,58]. Within the present study, a reduction in virus-induced mRNA synthesis of IP-10, IFNβ, IFNλ1, IFNλ2/3 and TRAIL of Calu-3 mono cell culture was observed, which was verified on the level of IP-10, IFNβ, IFNλ1 and IFNλ2/3 protein secretion. In the more complex chip model, a LASAG-mediated reduction in IP-10 and IFNλ1 protein secretion was observed. The SARS-CoV-2-induced IL-6 protein secretion was not affected in the presence of LASAG.

The difference in the LASAG-mediated effect on SARS-CoV-2-induced cytokine and chemokine expression might be due to the close proximity of epithelial cells, endothelial cells and PBMCs. Thus, it might be that the interplay of these cells provokes differences in the cytokine expression. Furthermore, the prolonged growth of cells in this model system might be a reason for lower infection and replication compared to monocultured cells and probably leads to changes in infection-related effects.

The different effects of LASAG in vitro and in vivo are probably due to the difference in concentration. Millimolar amounts of LASAG, which were used in in vitro experiments to study coronavirus but also IAV infection [34,36], cannot be achieved in vivo. More than 3 g/L would be required to reach a concentration of 10 mM in the blood, but this is toxic [59]. After intravenous administration of 500 mg ASA i.v. the highest plasma concentration is 54.25 mg/L and after oral administration 4.84 mg/L [34,60]. Indeed, we encountered some toxicity of LASAG on the endothelial layer of the chip model, which was not observed in mono cell culture. Nonetheless, the application directly to the lung via inhalation may be considered, since a study on the treatment of asthma with LASAG showed that higher concentrations can be achieved this way without causing toxicity [61]. Further investigations are required to determine if patients might benefit from the anti-viral and anti-inflammatory effects of LASAG.

In addition, it might be considered to increase the antiviral effect of LASAG by combinatory treatment with antiviral substances, such as remdesivir. Combination therapies are standard for the treatment of viral infections due to a better efficacy, decreased toxicity and the prevention of resistance emergence compared to single antivirals [45]. With the help of the webtool Syngeryfinder 2.0, several pharmacological studies were carried out to find useful drug combinations that could be repurposed to treat COVID-19 [62,63,64]. Our data clearly indicate an additive effect on SARS-CoV-2 inhibition by combining LASAG with remdesivir in vitro.

In severe COVID-19 cases, a dysfunction of the lungs and other organs is linked to vascular injuries and thrombosis and several blood parameters are activated during this process [65,66]. Among those, TF is the primary initiator of blood coagulation and is activated after vessel injury. Aberrant TF expression can induce intravascular thrombosis [67]. Interestingly, TF was shown to be activated during infections with Herpes simplex virus, HIV and Ebola as well as SARS-CoV-2 infection [68,69,70,71]. In primates, ASA-mediated TF inhibition did not result in increased bleeding [72], and in pigs, TF inhibition with recombinant TF-pathway inhibitor and ASA prevented acute thrombosis by decreasing thrombus areas without further bleeding complications [73]. Clinical studies showed that TF pro-coagulant activity declined after combined treatment with a coagulation inhibitor (clopidogrel) and ASA [74] and that a combined treatment with ASA and other antiplatelet-acting drugs reduced TF levels and thrombin generation in patients with angina pectoris [75]. Similarly, TF inhibition has been discussed as a potential therapeutic strategy in COVID-19 patients in several studies [76,77,78]. Our data showed a significant increase in TF expression in the endothelial layer of the chip model after SARS-CoV-2 infection of the epithelial layer. Since the endothelial layer was not infected, these results indicated an indirect deregulation of hemostasis probably due to SARS-CoV-2-induced cell responses.

The two coagulation factors tPA and PAI-1 are part of the fibrinolytic system and—in balance—dissolve blood clots and prevent blood clotting [79]. Elevated tPA and PAI-1 levels were found in patients hospitalized with COVID-19 and these high levels were associated with a worse respiratory status [80]. In our studies, we showed a strong induction of PAI-1 after SARS-CoV-2 infection in mono cell culture followed by a LASAG-dependent significant reduction. The tPA was not affected in mono cell culture studies. In the chip model, both tPA and PAI-1 were not induced but the significant reduction in tPA of the LASAG-treated endothelial cells correlates with very early studies on ASA which show an inhibition of tPA activity by ASA [81].

Taken together, numerous beneficial mechanisms of the action of ASA have been reported during COVID-19, although the negative side effects cannot be ignored. Our own in vitro data show a LASAG-mediated reduction in viral titers, SARS-CoV-2-induced cytokines, chemokines and coagulation factors, when used at high concentrations. Admittedly, this treatment also resulted in the disturbance of the endothelial layer in the chip model, which was independent of the SARS-CoV-2 infection, and maybe due to limited growth areas or medium exchange. However, LASAG-mediated cell damage was neither observed in mono cell culture for the endothelial cells nor for epithelial cells. Nevertheless, such millimolar concentrations of LASAG cannot be achieved in vivo, but were always used in cell culture experiments to demonstrate the antiviral and anti-inflammatory effects of ASA and LASAG during viral infections [34,36].

Since aerosolized LASAG was already used for the treatment of influenza in hospitalized patients [31] and based on the fact that LASAG was able to reduce SARS-CoV-2 load, SARS-CoV-2-induced pro-inflammatory cytokines and thrombotic factors, this substance seems to be promising for future studies against COVID-19. In particular, combinations with antiviral substances, such as remdesivir that could increase the effect and reduce toxicity, might be considered.

## 4. Materials and Methods

### 4.1. Cell Culture, Cytotoxicity and Virus Infection

For mono cell culture, Calu-3 (ATCC Cat. No. HTB-55; provided from the laboratory of Stephan Ludwig, Münster, Germany) and Vero-76 (ATCC Cat. No. 1587, cell stock of the former Institute for Antiviral Therapy, Jena, Germany) cells were cultivated in Dulbecco’s Modified Eagle’s Medium (DMEM, high Glucose, Sigma-Aldrich, Taufkirchen, Germany) supplemented with 10% fetal calf serum (FCS, PAN-Biotech, Aidenbach, Germany). Aspirin^®^ i.v. 500 mg (Bayer vital, Leverkusen, Germany), referred to as LASAG, was solved in DMEM/FCS to obtain a 200 mM stock solution and further diluted in DMEM/FCS to the final concentrations. Remdesivir (GS-5734) (Selleckchem, Planegg, Germany) was solved in DMSO to obtain a 10 mM stock solution and further diluted in DMEM/FCS to the final concentrations.

For determining the cytotoxicity of drugs, Calu-3 and Vero-76 cells were seeded in 24-well plates and treated with different concentrations for 24 h at 37 °C. After that, the viable cell number was counted on a Countess^TM^ II (Invitrogen, Dreieich, Germany).

Mono cell culture infection was performed as follows: cells were washed with phosphate-buffered saline (PBS, Biozym, Hessisch Oldendorf, Germany) and pre-treated for 1 h with LASAG, remdesivir or solvent controls at the indicated concentrations in DMEM/FCS. The cells were either left uninfected or were infected with SARS-CoV-2 at the indicated MOI for 2 h in DMEM/FCS supplemented with the indicated pharmaceutical substances and concentrations. After infection, the medium was removed, cells were washed with PBS and further cultivated in fresh DMEM/FCS supplemented with the substances at 37 °C and 5% CO_2_.

The BioChips (BC-002) were purchased commercially from the Dynamic42 GmbH (Jena, Germany). The BioChips were equipped with epithelial and endothelial cells as well as PBMCs as described previously [44,82]. For use in the chip model, the Calu-3 cells were cultured in RPMI 1640 (Sigma-Aldrich, Taufkirchen, Germany) supplemented with 10% FCS and the HUVECs in endothelial cell growth medium (EC, Promocell, Heidelberg, Germany) including supplement mix (Promocell, Heidelberg, Germany). HUVECs were isolated from anonymously acquired human umbilical cords according to the Declaration of Helsinki, “Ethical Principles for Medical Research Involving Human Subjects” (1964). LASAG was solved in RPMI1640 to obtain a 200 mM stock solution. Furthermore, LASAG was diluted in RPMI1640/FCS or EC/supplements for incubation in the epithelial chamber or the endothelial chamber, respectively. For the infection of the chip model, the membranes were washed with PBS and pre-treated for 1 h with 10 mM LASAG in both chambers. The epithelial chamber was left uninfected or infected with SARS-CoV-2 (MOI 1) in RPMI1640 supplemented with 0.2% autologous human serum, 1 mM MgCl_2_, 0.9 mM CaCl_2_. After 3 h incubation, the membranes were washed and supplemented with fresh medium containing LASAG at 37 °C and 5% CO_2_.

### 4.2. Viruses and Standard Plaque Assay

All SARS-CoV-2 experiments were performed in a laboratory of biosafety level 3.

SARS-CoV-2 isolates from respiratory specimens of four different patients: Alpha variants SARS-CoV-2/hu/Germany/Jena-vi005159/2020 [isolate 5159] (MW633322.1), SARS-CoV-2/hu/Germany/Jena-vi005587/2020 [isolate 5587] (MW633323.1) and SARS-CoV-2/hu/Germany/Jenavi005588/2020 [isolate 5588] (MW633324.1) [44] and Delta variant SARS-CoV-2/hu/Germany/Jena-0114749/2021 [isolate 4749] (ON650061) were used for infection experiments. The isolate 5159 belongs to the B.1-lineage, the isolates 5587 and 5588 to the B.55-lineage and the isolate 4749 to the AY.126-lineage.

For plaque assay, Vero-76 cells were seeded in 6-well plates and infected with serial dilutions of the supernatants in PBS supplemented with 1 mM MgCl_2_, 0.9 mM CaCl_2_, 0.2% BSA and 100 U mL^−1^ Pen/Strep (Sigma-Aldrich, Taufkirchen, Germany) for 90 min at 37 °C. After aspiration, the cells were incubated with 2 mL MEM supplemented with 0.9% agar (Oxoid, Wesel, Germany), 0.01% DEAE-Dextran (Pharmacia Biotech, Freiburg im Breisgau, Germany), 0.2% BSA and 0.2% NaHCO_3_ (Sigma-Aldrich, Taufkirchen, Germany) at 37 °C and 5% CO_2_ for 3 days. To visualize the plaques, a staining with neutral red solution (Sigma-Aldrich, Taufkirchen, Germany) in PBS was performed and the number of infectious particles (pfu mL^−1^) was determined by counting.

### 4.3. Immunfluorescence Microscopy

For immunofluorescence microscopy studies, Calu-3 cells were cultivated in 24-well plates supplemented with glass slides and infected as described above. At 8 h or 24 h p.i., the slides were fixed for 30 min in 4% paraformaldehyde (PFA, Sigma-Aldrich, Germany) at 37 °C and permeabilized with 0.1% Triton-X 100 (Sigma-Aldrich, Taufkirchen, Germany) for 15 min.

The human chip model membranes were fixed for 30 min with 4% PFA at 37 °C. The membrane was removed from the chip and cut in half to analyze either the epithelial or the endothelial side and permeabilized in PBS supplemented with 0.1% Saponin (Sigma-Aldrich, Taufkirchen, Germany) and 3% goat serum (Invitrogen, Dreieich, Germany) for 1 h at room temperature.

A mouse anti-SARS-CoV-2 spike IgG monoclonal antibody (GeneTex, Hsinchu City, Taiwan; no. GTX632604) and an Alexa Fluor 488-conjugated goat anti-mouse IgG polyclonal antibody (Dianova, Hamburg, Germany; no. 115-545-146) were used to visualize the SARS-CoV-2 infection. Nuclei were visualized by BisBenzimide H 33342 trihydrochloride (Hoechst 33342) (Merck, Darmstadt, Germany; no. 14533). Rabbit anti-E-Cadherin IgG monoclonal (Cell Signaling; no. 3195S) or rabbit anti-VE-Cadherin polyclonal antibodies (Cell Signaling; no. 2158S) and Cy5 goat anti-rabbit IgG polyclonal antibodies (Dianova, Hamburg, Germany; no. 111-175-144) were used to stain cell borders of Calu-3 or HUVEC cells, respectively. On both the slides and membranes of the chip model, primary antibodies were added (1:100) and incubated overnight at 4 °C. The secondary antibodies (1:100) and Hoechst 33342 (1:1000) were added and incubated for 1 h at room temperature in the dark. Slides and membranes were mounted with fluorescence mounting medium (Dako; no. S3023).

The fluorescent images were acquired using an Axio Observer.Z1 microscope (Zeiss, Jena, Germany) with Plan Apochromat 20×/0.8 objective (Zeiss, Jena, Germany), ApoTome.2 (Zeiss, Jena, Germany) and Axiocam 503 mono (Zeiss, Jena, Germany) and the software Zen 2.6 (blue edition; Zeiss, Jena, Germany). Apotome defolding with phase error correction and deconvolution and Z-stack merging with maximum intensity projection was done carried out with the software Zen 2.6 as well.

### 4.4. Western Blot Analysis

For Western blotting, cells were lysed with Triton lysis buffer (20 mM Tris-HCl, pH 7.4; 137 mM NaCl; 10% glycerol; 1% Triton X-100; 2 mM EDTA; 50 mM sodium glycerophosphate; 20 mM sodium pyrophosphate; 5 mg mL^−1^ aprotinin; 5 µg mL^−1^ leupeptin; 1 mM sodium vanadate; and 5 mM benzamidine) for at least 30 min at 4 °C. Cell lysates were cleared by centrifugation, supplemented with 5xLaemmli buffer (10% SDS, 50% glycerol, 25% β-mercaptoethanol, 0.02% bromophenol blue, 312 mM Tris, pH 6.8) and boiled for 10 min at 95 °C. Equal volumes were loaded on 10% SDS-PAGE and blotted on 0.2 µm Nitrocellulose membranes.

For the detection of SARS-CoV-2 spike protein, a SARS-CoV-2 spike S2 antibody (Sino Biological, Eschborn, Germany; no. 40590-T62) was used. Further antibodies against phosphorylated STAT1 (pY701) (BD Bioscience, Heidelberg, Germany; no. 612133), STAT1 (BD Bioscience, Heidelberg, Germany; no. 610115), PARP1 (BD Bioscience, Heidelberg, Germany; no. 611039) and ERK2 (D-2) (Santa Cruz, Heidelberg, Germany; no. sc1647) were used.

The membranes were developed in a Fusion©FX6.Edge (Vilber Lourmat, Eberhardzell, Germany) with Fusion© software Evolution-Capt (Vilber Lourmat, Eberhardzell, Germany) using ECL Western blotting substrate (Pierce^TM^, Thermo, Dreieich, Germany). For the quantification of the signals, the webtool ImageJ.JS (https://ij.imjoy.io/) was used (access dates: Figure 2a,b,e: 4 April 2022; Figure 2c,d: 6 June 2022).

### 4.5. Detection of mRNA Expression by Using qRT-PCR

RNA isolation was performed using the RNeasy mini kit (Qiagen, Hilden, Germany) according to the manufacturer’s protocol. The cells were lysed and scraped with 350 µL RLT lysis buffer and the RNA was eluted at the end in 30 µL RNase-free water. A NanoDrop spectrophotometer ND-1000 (Peqlab, Erlangen, Germany) was used for measuring the RNA concentration.

QuantiNova reverse transcription kit (Qiagen, Hilden, Germany) was used for cDNA synthesis. In total, 500 ng RNA was diluted to a total volume of 13 µL with RNase-free water and together with 2 µL gDNA removal mix incubated at 45 °C for 2 min. After cooling, 5 µL RT master mix (4 µL reverse transcription mix and 1 µL reverse transcription enzyme) was added to each sample and incubated at 25 °C for 3 min, at 45 °C for 10 min and inactivated at 85 °C for 5 min.

QuantiNova SYBR green PCR kit (Qiagen, Hilden, Germany) was used for the qRT-PCRs. In total, 19 µL master mix (10 µL 2xSYBR green, 1.5 µL 10 µM forward primer, 1.5 µL 10 µM reverse primer, 6 µL RNase-free water) and 1 µL cDNA were multiplied in a 72-well RotorGene (Qiagen, Hilden, Germany) in the following cycling conditions: 2 min 95 °C, 40 cycles of 5 s 95 °C and 10 s 60 °C. The melting curves were obtained by a stepwise temperature increase (1 °C every 5 s) from 60 °C to 95 °C. The Cq-values were normalized to GAPDH and compared to the SARS-CoV-2-infected samples without drugs.

For primer sequences, see Table 1.

### 4.6. Flow Cytometry Analyses

The flow cytometry analyses were performed using the LEGENDplex^TM^ assays (Biolegend, Amsterdam, Netherlands) according to the manufacturer’s protocol. In total, 25 µL of supernatant of Calu-3 cells or the different chambers of the chip model were incubated with the premixed beads (Human Anti-Virus Response Panel, Cat. 740349; Human Thrombosis Panel, Cat. 740891) overnight at room temperature on a plate shaker. The detection antibodies were incubated for 1 h and afterwards the SA-PE was added and further incubated for 30 min at room temperature on a plate shaker. The samples were fixed for 30 min in 4% PFA and after washing directly measured on an Accuri C6 Plus flow cytometer (BD Bioscience, Heidelberg, Germany). The data analysis was performed using the LEGENDplex^TM^ webtool powered by QOGNIT (San Carlos, CA, USA) (https://legendplex.qognit.com/, access dates: Figure 3b: 24 September 2021; Figure 3c: 20 October 2021; Figure 4b,c: 31 March 2021; Appendix A: 24 September 2021).

### 4.7. Statistical Analysis

Statistical analyses were performed in GraphPad Prism 8, the methods used are described in the figure legends.

## Figures and Tables

**Figure 1 ijms-23-06880-f001:**
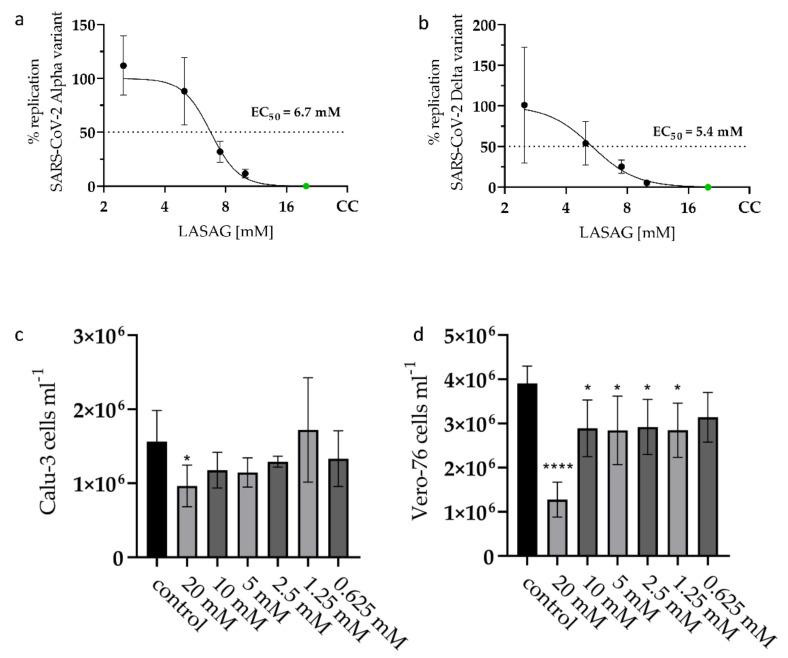
LASAG inhibits SARS-CoV-2 replication at non-toxic concentrations in vitro. (**a**,**b**) Calu-3 cells were left untreated or were incubated with the indicated concentrations of LASAG for 1 h. Subsequently, cells were infected with the SARS-CoV-2 variants ((**a**) Alpha (isolate 5159); (**b**) Delta (isolate 4749)) at 0.5 MOI (=multiplicities of infection) in the absence and presence of LASAG for 2 h. After a medium change, cells were further incubated in the absence and presence of LASAG. At 24 h post infection (p.i.), progeny virus titers were determined in the supernatant by standard plaque assay. The virus titers of control-infected cells were arbitrarily set to 100%. Shown are the means (±SD) of three independent experiments including two biological samples. EC_50_ were calculated by a 4 parameter nonlinear fit in Graphpad prism. (**c**,**d**) Calu-3 (**c**) and Vero-76 (**d**) cells were cultivated for 24 h in the absence and presence of LASAG. The cell viability was evaluated by cell counting. Shown are the means (±SD) of viable cells of three independent experiments including two biological samples. Statistical significance was analyzed by ordinary one-way ANOVA followed by Dunnett’s multiple comparisons test (*, *p* < 0.0332; ****, *p* < 0.0001).

**Figure 2 ijms-23-06880-f002:**
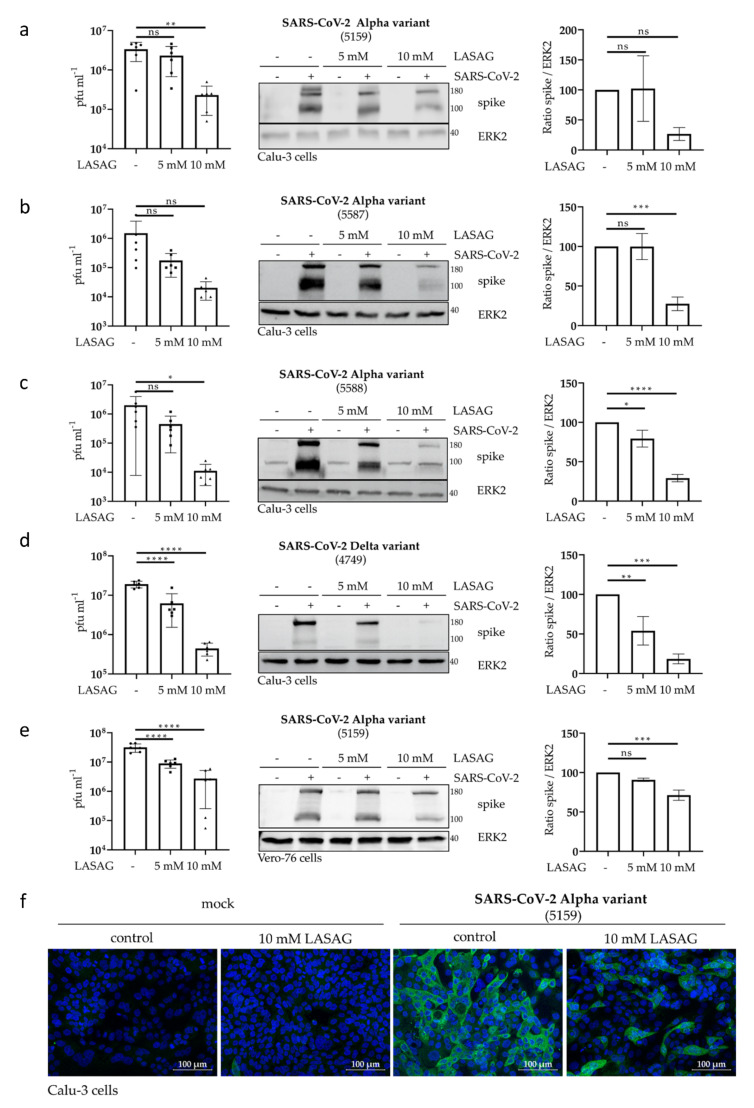
In the presence of LASAG, SARS-CoV-2 replication of different variants is significantly reduced in vitro. Calu-3 (**a**–**d**,**f**) and Vero-76 (**e**) cells were left untreated or were incubated with the indicated concentrations of LASAG for 1 h. Subsequently, cells were left uninfected (mock) or were infected with the SARS-CoV-2 variants (Alpha variants: isolate 5159 (**a**,**e**,**f**), isolate 5587 (**b**), isolate 5588 (**c**); Delta variant: isolate 4749 (**d**)) at 0.5 MOI in the absence and presence of LASAG for 2 h. After a medium change, cells were further incubated in the absence and presence of LASAG. At 24 h p.i., the analyses were performed. (**a**–**e**; left panel) Progeny virus titers were determined in the supernatant by standard plaque assay. Shown are the means (±SD) of plaque forming units (pfu) mL^−1^ of three independent experiments including two biological samples. (**a**–**e**; middle panel) Total cell lysates were used to investigate the viral spike protein expression by Western blot analysis. ERK2 served as the loading control. Shown are representative example blots of three independent experiments. (**a**–**e**; right panel) The spike protein signals were analyzed with ImageJ and normalized to the ERK2 signals. Shown are the means (±SD) of three independent experiments. Untreated, infected samples were arbitrarily set to 100%. (**f**) In immunofluorescence microscopy, SARS-CoV-2 spike protein was detected by a spike-specific antibody and an Alexa Fluor 488-conjugated goat anti-mouse IgG antibody (green). The nuclei were stained with Hoechst 33342 (blue). Immunofluorescence microscopy was acquired by use of the Axio Observer.Z1 instrument (Zeiss) with a ×200 magnification. Scale bars represent 100 µm. Shown are the representative example pictures of three independent experiments. (**a**–**e**) Statistical significance was analyzed by ordinary one-way ANOVA followed by Dunnett’s multiple comparisons test (ns, non-significant; *, *p* < 0.0332; **, *p* < 0.0021; ***, *p* < 0.0002; ****, *p* < 0.0001).

**Figure 3 ijms-23-06880-f003:**
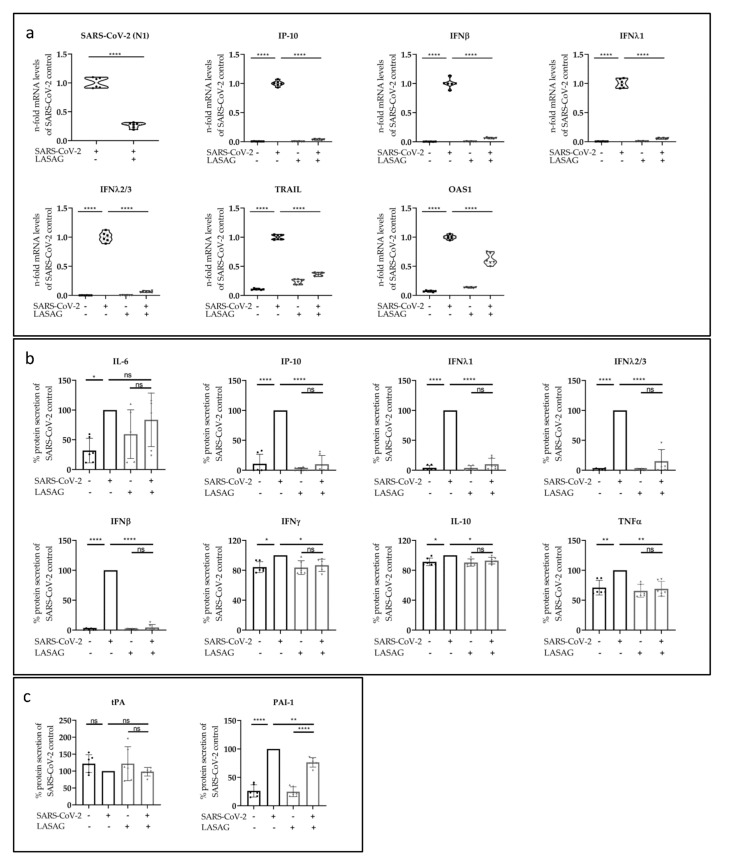
LASAG affects SARS-CoV-2-induced mRNA synthesis and cytokine production in vitro. Calu-3 cells were left untreated or were incubated with 10 mM of LASAG for 1 h. Subsequently, cells were left uninfected or were infected with the SARS-CoV-2 Alpha variant (isolate 5159) at 0.5 MOI in the absence and presence of LASAG for 2 h. After a medium change cells were further incubated in the absence and presence of LASAG. At 24 h p.i., the analyses were performed. (**a**) RNA was isolated and mRNA expression was investigated by qRT-PCR. Data represent three independent experiments including two biological replicates. The relative mRNA levels are shown as the 2^−∆∆C(T)^ of untreated, infected samples. (**b**,**c**) Supernatants were collected and different factors were analyzed by flow cytometry (LEGENDplex^TM^): (**b**) IL-6, IP-10, IFNλ1, IFNλ2/3, IFNβ, IFNγ, IL-10 and TNFα and (**c**) tPA and PAI-1. Means (±SD) of three independent experiments including two biological replicates are shown. The mean levels of untreated, infected samples were arbitrarily set to 100%. (**a**) The 2^−∆∆C(T)^-values and (**b**,**c**) the levels after normalization were analyzed by ordinary one-way ANOVA followed by Dunnett’s multiple comparison test (ns, non-significant; *, *p* < 0.0332; **, *p* < 0.0021; ****, *p* < 0.0001).

**Figure 4 ijms-23-06880-f004:**
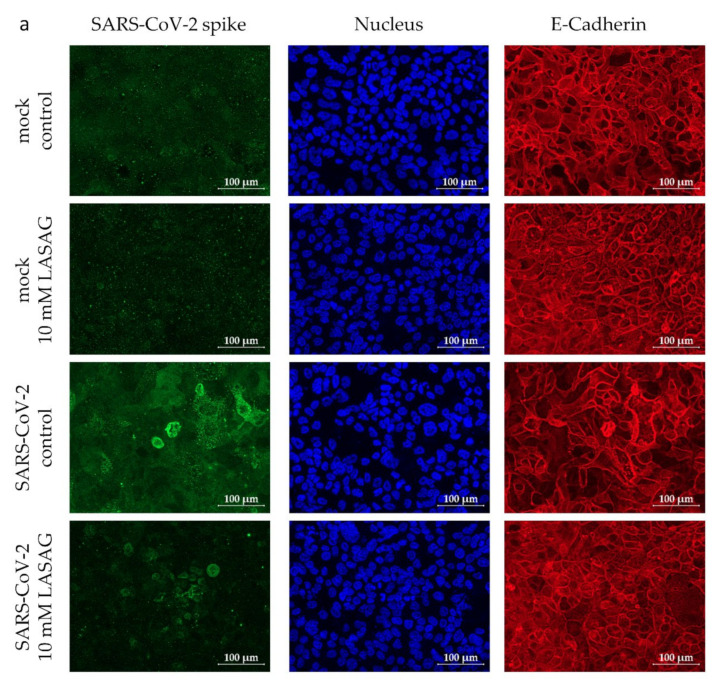
LASAG treatment results in the reduction in SARS-CoV-2-induced cytokine synthesis in a chip model. Both the epithelial and endothelial chambers were left untreated or were incubated with 10 mM of LASAG for 1 h. Subsequently, the chambers were left uninfected (mock) or were infected with the SARS-CoV-2 Alpha variant (isolate 5159) at 1 MOI in the absence and presence of LASAG for 3 h. After a medium change, cells were further incubated in the absence and presence of LASAG. At 28 h p.i., the analyses were conducted. (**a**) Immunofluorescence staining of the epithelial layer was performed and the SARS-CoV-2 spike protein was visualized via a spike-specific antibody and an Alexa Fluor 488-conjugated goat-anti mouse IgG antibody (green). The nuclei were stained with Hoechst 33342 (blue). The E-Cadherin of the epithelial cells was visualized by anti-E-Cadherin-specific antibody and a Cy5-conjugated goat-anti rabbit IgG antibody (red). Scale bars represent 100 µm. (**b**,**c**) Different factors of supernatants of (**b**) the epithelial and (**c**) the endothelial chamber were analyzed by flow cytometry. Means (±SD) of three independent experiments are shown. Levels of untreated, infected samples were arbitrarily set to 100%. After normalization, ordinary one-way ANOVA followed by Dunnett’s multiple comparisons test were performed (ns, non-significant; *, *p* < 0.0332).

**Figure 5 ijms-23-06880-f005:**
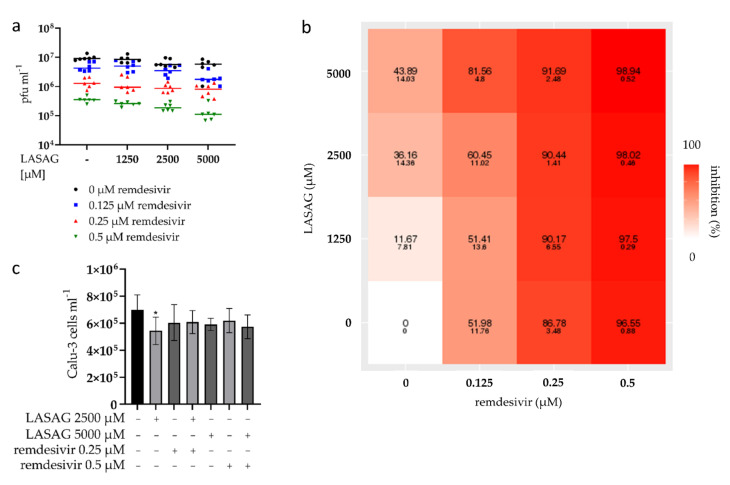
LASAG shows additive effects on SARS-CoV-2 replication in vitro when combined with remdesivir. Calu-3 cells were left untreated or were incubated with the LASAG–remdesivir combinations for 1 h. Subsequently, cells were infected with the SARS-CoV-2 Alpha variant (isolate 5159) at 0.5 MOI in the absence and presence of the LASAG–remdesivir combinations for 2 h. After a medium change, cells were further incubated in the absence and presence of the LASAG–remdesivir combinations. (**a**) Progeny virus particles were measured in the supernatant by standard plaque assay at 24 h p.i. Shown are the means of pfu mL^−1^ of three independent experiments including two biological samples. (**b**) The calculated inhibitions (and SD) were used to evaluate the interactions of LASAG and remdesivir and are depicted in a heatmap. (**c**) The highest concentrations of both substances used in combination show no effect on cell growth, which was determined by cell counting after 24 h of incubation. Statistical significance was analyzed by ordinary one-way ANOVA followed by Dunnett’s multiple comparisons test (*, *p* < 0.0332).

**Table 1 ijms-23-06880-t001:** Primer sequences.

Name	Sequence 5′-3′
GAPDH-fwd	CTCTGCTCCTCCTGTTCGAC
GAPDH-rev	CAATACGACCAAATCCGTTGAC
2019_nCoV_N1-fwd	GACCCCAAAATCAGCGAAAT
2019_nCoV_N1-rev	TCTGGTTACTGCCAGTTGAATCTG
IP-10-fwd	CCAGAATCGAAGGCCATCAA
IP-10-rev	TTTCCTTGCTAACTGCTTTCAG
IFN-β-fwd	ATGACCAACAAGTGTCTCCTCC
IFN-β-rev	GGAATCCAAGCAAGTTGTAGCTC
IFN-λ1-fwd	CGCCTTGGAAGAGTCACTCA
IFN-λ1-rev	GAAGCCTCAGGTCCCAATTC
IFN-λ2/3-fwd	AGTTCCGGGCCTGTATCCAG
IFN-λ2/3-rev	GAGCCGGTACAGCCAATGGT
TRAIL-fwd	GTCTCTCTGTGTGGCTGTAACTTACG
TRAIL-rev	AAACAAGCAATGCCACTTTTGG
OAS1-fwd	GATCTCAGAAATACCCCAGCCA
OAS1-rev	AGCTACCTCGGAAGCACCTT

## Data Availability

The data presented in this study are available in the present article and Appendix A.

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
