# Peer review of "D,L-Lysine-Acetylsalicylate + Glycine (LASAG) Reduces SARS-CoV-2 Replication and Shows an Additive Effect with Remdesivir"

_ijms, 2022, doi:10.3390/ijms23136880_

Round 1

Reviewer 1 Report

The authors made all the necessary changes and carried out additional research. The article can be accepted for publication.

Reviewer 2 Report

The revised MS entitled “D,L-lysine-acetylsalicylate + glycine (LASAG) reduces SARS-CoV-2 replication and shows an additive effect with remdesivir” has relieved my concerns and is acceptable now.

This manuscript is a resubmission of an earlier submission. The following is a list of the peer review reports and author responses from that submission.

Round 1

Reviewer 1 Report

LASAG has been demonstrated to have antiviral potential towards human coronaviruses before. Besides, LASAG is widely recognized to have anti-inflammatory and anti-thrombotic properties. In this study, the authors aimed to investigate the effects of LASAG during SARS-CoV-2 infections in vitro. The presented evidence shows the clear antiviral and modest anti-inflammatory and anti-thrombotic effects. However, some concerns are needed to be relieved here.

  1. The resolution of figures, particularly statistic graphs, is not good. Please provide high-quality figures.
  2. The signals for spike proteins are too low in Figure 1c. Please try to improve it.
  3. (Fig 1a-1d) Since there are two major bands shown in anti-spike immunoblotting, please label the molecular weight using a protein marker on the membrane and describe the difference between these two bands.
  4. This study only used the alpha variant. Thus far, there are five variants of concern. Please test the antiviral effect of LASAG on other variants (at least one variant).
  5. Fig 2a (MOI=0.5) is to check RNA expression and Fig 2b & 2c (MOI=1) is to check the protein expression in the supernatants. It is good to show both data. However, the MOI should be consistent. Please redo the experiment with the same MOI.
  6. (Figure 4) Please monitor the cytotoxicity of the combinatory treatment of LASAG and remdesivir.
  7. Please discuss why LASLG causes different effects on SARS-CoV-2 induced cytokine and chemokine in monoculture Calu3 cells (Fig. 2b) and in the chip model (Fig3b).

Reviewer 2 Report

The work presented here is of great importance for the search for new effective drugs in the fight against the coronavirus infection that is sweeping the world. The authors studied the effects of aspirin and its salts under in vitro conditions and tried to explain the mechanism of action.

The main editorial comments are as follows:

  1. Figures divided into several parts look very bad, they should be redone. For example, Figure 1 is placed on both pp. 1, and on p. 2. Maybe we should separate the figures and put them in the text; it would be much more convenient for the reader. The caption to the figure on page 1 and the continuation on page 2 is not the best option. The same goes for figures 2 and 3. This needs to be redone.
  2. The list of references does not comply with the rules of the journal; the authors should carefully study the requirements for design.
  3. It should be clarified to which phylogenetic lineage the SARS-CoV-2 virus strains studied in this paper belong.
  4. It should be clarified what viral load was used in the in vitro experiments and how many cytopathic doses (TCD50) were added to the cells.
  5. It would be good if the authors gave IC50 data for aspirin and its analogues.
  6. In the experiments to study the synergistic effect of aspirin and remdesevir (Figure 4b), the dose of remdesevir in micromoles and the dose of LASAG in millimoles. It would have been more competent to bring everything to the same concentrations.
  7. Perhaps it would make sense to bring the structural formulas of the compounds under study. Although everyone knows what aspirin is, the second agent LASAG would be better clarified.